# Transcriptomics-Based Identification of Genes Related to Tapetum Degradation and Microspore Development in Lily

**DOI:** 10.3390/genes13020366

**Published:** 2022-02-17

**Authors:** Juanjuan Sui, Wenjie Jia, Yin Xin, Yuanyuan Zhang

**Affiliations:** 1Department of Biology, Biology and Food Engineering College, Fuyang Normal University, Fuyang 236037, China; suijuanjuan@163.com; 2Beijing Key Laboratory of Development and Quality Control of Ornamental Crops, Department of Ornamental Horticulture and Landscape Architecture, China Agricultural University, Beijing 100193, China; jiawenjie917@aliyun.com (W.J.); xy161354@163.com (Y.X.); 3Flower Research Institute, Yunnan Academy of Agricultural Sciences, Kunming 650205, China; 4College of Biological Sciences, China Agricultural University, Beijing 100193, China

**Keywords:** lily, pollen grains, tapetum degradation, microspore development, transcriptome

## Abstract

Lily is a popular and economically ornamental crop around the world. However, its high production of pollen grains causes serious problems to consumers, including allergies and staining of clothes. During anther development, the tapetum is a crucial step for pollen formation and microspore release. Therefore, it is important to understand the mechanism of tapetum degradation and microspore development in lily where free pollen contamination occurs. Here, we used the cut lily cultivar ‘Siberia’ to characterize the process of tapetum degradation through the use of cytology and transcriptomic methods. The cytological observation indicated that, as the lily buds developed from 4 cm (Lo 4 cm) to 8 cm (Lo 8 cm), the tapetum completed the degradation process and the microspores matured. Furthermore, by comparing the transcriptome profiling among three developmental stages (Lo 4 cm, Lo 6 cm and Lo 8 cm), we identified 27 differentially expressed genes. These 27 genes were classed into 4 groups by function, namely, cell division and expansion, cell-wall morphogenesis, transcription factors, LRR-RLK (leucine-rich repeat receptor-like kinases), plant hormone biosynthesis and transduction. Quantitative real-time PCR was performed as validation of the transcriptome data. These selected genes are candidate genes for the tapetum degradation and microspore development of lily and our work provides a theoretical basis for breeding new lily cultivars without pollen.

## 1. Introduction

Lily is one of the most important cut flowers in the world due to its beautiful blossoms and pleasant fragrance [1,2]. However, the pollens released from anthers cause environmental pollution leading to consumer issues, such as pollen allergies and staining of clothes and petals [1,3,4]. Although doubled lily varieties have no stamens and grow additional petals instead, consumers still prefer single lilies, such as popular cultivars ‘Sorbonne’ and ‘Siberia’, which still maintain their anthers and stamens [1,4]. Therefore, a sterile single lily would be the ideal cultivar for consumers.

A mature anther is composed of four somatic layers [5,6]. From the outside to the inside, they are the epidermis, the endothecium, the middle layer and the tapetum [5,6]. The tapetum is the innermost layer to the anther chamber and it plays an essential role in microspore development. The tapetum provides nutrition for microspores, degrades callose and participates in pollen exine and coating formation [5,6,7]. Previous studies have shown that there are several genes involved in tapetum degradation and microspore development in model plants, e.g., Arabidopsis and rice. In the *myb33/myb65* mutant, tapetum cells vacuolated at the pollen mother-cell stage, leading to microspore degradation before meiosis [8]. In the *dyt1* (*dysfunctional tapetum 1*) mutant, although normal tapetum cells could form, the tapetum cells were non-functional [9]. At the 6th and 7th stages of anther development in Arabidopsis, the tapetal cells were abnormally hypertrophic and filled the chamber, squeezing the developing tetrad, which led to the early degradation of microspores [9]. Similarly, in the *tdf1* (*defective in meristem development and function 1*) mutant, the tapetum showed severe vacuolization and irregular division, followed by abnormal hypertrophy, leading to microspore degeneration [10]. In the *ams* (*aborted microspores*) mutant, the microspore mother cells experienced normal meiosis, but, at a later stage, the tapetum vacuolated and hypertrophied abnormally after the microspore was released from the tetrad, which led to the early degradation of microspores [11,12,13]. In the *ms188* (*male sterile 188*) mutant, pollen mother cells developed normally in the early stage; however, the tapetum could not develop into normal secretory cells at a later stage, which also led to the early degradation of microspores [6,14,15]. In the *ms1* mutant, the tapetum cells vacuolated, which led to the dysfunction of the tapetum and the degradation of microspores [16,17]. Therefore, abnormal tapetum development can result in pollen abortion.

Male sterility reduces the work of artificial emasculation and solves the problem of pollen contamination and allergy in ornamental plants, such as lily [3]. Although much progress has been made in the field of anther development, the mechanism largely remains to be investigated [13]. More recently, RNA sequencing (RNA-Seq) has been used as a tool in ornamental plants to identify genes related to a variety of desirable traits [18].

To better understand the degradation of the tapetum and independent microspore releasing, we histologically analyzed lily anthers by paraffin sectioning. According to the sections analyzed, anthers at three different stages were selected for transcriptome sequencing, namely, Lo 4 cm (flower buds at 4 cm in length), Lo 6 cm and Lo 8 cm. The BGISEQ-500 sequencing platform was used to identify the differentially expressed genes (DEGs) related to tapetum degradation and independent microspore releasing in ‘Sibera’. Several important DEGs were identified, including some transcription factors (TFs). Our findings provide new insights into the potential molecular mechanism of tapetum degradation and microspore development. In addition, the candidate genes identified can be used for breeding new lily varieties without pollen contamination by genetic engineering.

## 2. Materials and Methods

### 2.1. Plant Materials and Treatments

Anthers collected from the Oriental Hybrid Lily cultivar ‘Siberia’ were used in the experiment. The lily was planted in greenhouses and grown under a photoperiod of 16 h of light and 8 h of darkness, at 22 °C, at Fuyang Normal University, Fuyang, Anhui Province, China. The anthers were collected from the flower buds in different sizes (1, 2, 3, 4, 5, 6, 7, 8, 9 and 10 cm in length). Each lily bud contained 6 anthers, one of which was fixed with FAA (Formalin-Aceto-Alcohol) solution and was used for histological analysis, while the remaining five were quickly frozen in liquid nitrogen and transferred to −80℃ to be stored for RNA sequencing and analysis of gene expression.

### 2.2. Lily Anther Histological Observation

Anthers embedded in paraffin were cut into 8 μm thick sections, corresponding to a size of about 0.5 mm × 0.5 mm × 0.5 mm. The processes were performed as previously described [4]. The sections were stained by safranine and fast green double dyeing. Section observation was performed under a light microscope (Nikon Eclipse E100; Tokyo, Japan).

### 2.3. RNA Extraction and Library Construction

The anthers were removed from the buds and total RNA was extracted from these samples using an RNAprep Pure Kit (DP441, Tiangen, Beijing, China). More than 20 μg of RNA was extracted from each group of samples and was used to construct cDNA libraries. The quality and quantity of the RNA samples were assessed using Qubit and Agilent 2100. A Clontech SMARTer PCR cDNA Synthesis Kit was used to construct the Iso-Seq library. The Iso-Seq library construction started with the mRNA purified with Oligo(dT) magnetic beads (NEB) and cDNA was synthesized by using the SMARTer PCR cDNA Synthesis Kit (Clontech, Mountain View, CA, USA) according to the Iso-Seq protocol (Pacific Biosciences, Menlo Park, CA, USA). Then, cDNA was amplified using KAPA HiFi DNA Polymerase (Roche, Basel, Switzerland) for 12 cycles followed by purification and size selection. The BluePippin Size Selection System protocol was performed by following the instructions described by Pacific Biosciences (PN 100-092-800-03). The insert size of the library was detected using Agilent 2100 to ensure the quality of libraries. Short-read RNA-Seq libraries were prepared using TruSeq stranded RNA Library Preparation kits and the supplied protocol (Illumina, San Diego, CA, USA) and sequenced on a HiSeq2500 platform using v2 sequencing chemistry to generate 2 × 150 paired-end reads.

### 2.4. Error Correction of PacBio Iso-Seq Full-Length cDNA Reads

The samples’ transcriptomes were sequenced using the Illumina NextSeq 500 (Illumina, San Diego, CA, USA) and Pacbio SMRT Sequel (Pacific Biosciences, Menlo Park, CA, USA) platforms for RNA-Seq and Iso-Seq, respectively. The Iso-Seq subread raw reads were firstly obtained from the raw SMRT raw long reads and used to generate the corrected circular consensus sequences (CCS) by SMRT Link v8.0 software (Pacific Biosciences, Menlo Park, CA, USA) (minLength, 50; maxLength, 15,000; minPasses, 1). Then, the sequences were divided into full-length sequences and non-full-length sequences according to the existence of the 5′-primers, 3′ primers and polyA tails. To improve consensus accuracy, full-length non-chimera (FLNC) reads were polished by Illumina short reads using the arrow and consensus transcripts were obtained using the isoform-level clustering algorithm ICE (iterative clustering for error correction). Finally, RNA-Seq short reads from the nine samples were employed to correct sequencing errors in the consensus transcripts using the program LoRDEC [19].

### 2.5. Gene Prediction and Annotation

Cogent (https://github.com/Magdoll/Cogent, accessed on 23 June 2021; v 2.1) was used to reconstruct isoforms into one or several full-length unique transcript model(s) based on the k-mer clustering and De Bruijn graph methods. It generates UniTransModels as distinct genes. GMAP [20] software was used to compare the transcripts to UniTransModels and SUPPA software was used to identify different alternative splicing types. Transcripts with different splicing sites were considered as alternative splicing isomers. Finally, highly similar isoforms were clustered and removed using CD-HIT [21]; the non-redundant representative transcripts were kept for further analyses.

To obtain comprehensive gene function information, the non-redundant representative transcripts were aligned to the nucleotide and protein databases, i.e., Nr, Pfam, KOG/COG, Swiss-Prot, KEGG and GO. Open reading fragments (ORFs) for each transcript were detected by TransDecoder with a minimum length of 100 amino acids (AAs). Where multiple ORFs were found in a single transcript, we used the first ORF in each transcript as the ‘representative’. In each transcript locus, the isoform containing the longest ORF was the ‘locus representative’ for functional annotation. Protein sequences were downloaded from Swiss-Prot (ftp://ftp.uniprot.org/pub/databases/uniprot/current_release/knowledgebase/complete/uniprot_sprot.fasta.gz, accessed on 23 June 2021) and NCBI NR (ftp://ftp.ncbi.nlm.nih.gov/blast/db/FASTA/, accessed on 23 June 2021). We then removed the records for proteins described as hypothetical or predicted in these well-characterized protein databases for the ‘locus representative’ protein sequences using BLASTp (NCBI-BLAST v2.2.28+) with the following parameters: outfmt, 6; max_target_seqs, 1; evalue 1e-5. Meanwhile, the Pfam database of 16,712 protein families was downloaded from (ftp://ftp.ebi.ac.uk/pub/databases/Pfam; release Pfam31.0, accessed on 23 June 2021). Pfam Scan was used to identify occurrences of the known protein domains documented in the Pfam database with the parameters ‘E 1e-5’. Transcription factors were predicted using iTAK v 4.0 [22]. The GO terms and pathways in KEGG for each gene were obtained from KOBAS using *Arabidopsis thaliana* protein sequences as reference [23].

### 2.6. Quantification of Transcripts Using Illumina Short Reads

Paired-end Illumina RNA-Seq reads from each tissue sample were trimmed to remove the adaptor sequences and low-quality bases using Trimmomatic (v0.32) [24] with the explicit option settings ILLUMINACLIP: adapters.fa: 2:30:10:1:true LEADING:3 TRAILING:3 SLIDINGWINDOW: 4:20 LEADING:3 TRAILING:3 MINLEN:25. Trimmed Illumina reads were aligned against the obtained transcripts from the Iso-Seq analysis by Bowtie2 [25]. RSEM (v1.3.3) [26] was used to calculate the FPKMs of all transcripts in each sample. Isoforms FPKMs were computed by summing the FPKMs of transcripts in each gene group. FPKMs mean fragments per kilo-base of exon per million fragments mapped, calculated based on the length of the fragment and read count mapped to this fragment.

DEGs were identified using the DESeq2 package with the threshold of logarithm transformation of fold change larger than 1 and *p*-value < 0.05 [27].

### 2.7. The ‘eFP-Seq’ Browser Analysis of DEGs

The Arabidopsis ‘eFP-Seq Browser’ tool is a useful tool to display transcriptome data visually through electronic fluorescent pictographic (eFP) images, which can present the relative expression level of genes in different stages of development (http://bar.utoronto.ca/efp/cgi-bin/efpWeb.cgi, accessed on 23 June 2021) [28]. In this study, tissue-specific data, including pollen development, were analyzed.

### 2.8. Quantitative Real-Time PCR

To further verify the gene expression levels of DEGs identified by RNA sequencing, quantitative real-time PCR was used to detect gene expression to further confirm the transcriptome results. All primers of genes are listed in Appendix A. The *Lo18S* rRNA was used as the internal reference for the quantitative expression analysis. Three biological replicates were performed. The 2^−ΔΔCT^ method was used for the data analyses [29].

## 3. Results

### 3.1. Lily Anthers Cytology Observation

The lily flower contains six anthers, each with four chambers. The anther chamber is the location of microspore development and maturation. When the anther is mature, it cracks naturally and pollen grains scatter from the chamber (Figure 1A). To track the developmental stage for tapetum degradation and independent microspore formation, we collected anthers of different sizes of flower buds (from 1 to 10 cm) and took out one of the six anthers from each flower for paraffin embedding and microscopic analysis. The histological results show that the four layers’ structure (tapetum, middle layer, endothecium and epidermal layer) gradually formed from 1 cm to 3 cm stage buds. The tapetal cells became long with a large nucleus and dense cytoplasm in the 3 cm buds, which is in accordance with previous results [4]. At the Lo 4cm bud stage, several changes occurred in the anther, i.e., (1) binuclear and polynuclear cells appeared; (2) cytoplasm condensed with cellular vacuolization; and (3) the pollen mother cells developed into dyads and tetrads (Figure 1B,C). From the Lo 4 cm to Lo 6 cm stages, the tapetal cytoplasm was further concentrated and independent microspores were released from the tetrads, suggesting callose degradation during this transition (Figure 1B,D) [4,14]. Thus, the transition from the Lo 4 cm to Lo 6 cm stages was the key stage of callose degradation. At the Lo 8cm stage, the whole cell structure of the tapetum could not be observed except for parts of the tangential wall (Figure 1B,E). The tapetum provides protein, lipid and other nutrients for microspores development; consequentially, it was completely degraded before pollens were mature [5]. We observed microspore development from binuclear cells to mature pollen from the Lo 4 cm to Lo 8 cm stages. Taken together, alongside tapetum degradation, microspores developed and matured.

### 3.2. Overall Analysis of Transcriptome Sequencing and Gene Annotation

To obtain accurately expressed transcripts from the samples, we conducted Iso-Seq of pooled samples and RNA-Seq of nine samples from three development stages with three biological replicates. After transcript assembly and sequence correction, a total of 35,213 transcripts (17,065 genes) were identified and principal compound analysis (PCA) indicated high agreement among biological replicates (Figure 2A and Appendix A). The maximum length of unigenes was nearly 8.9 kb and the minimum length was 87 bp. N50 and N90 are important parameters reflecting the quality of transcript assembly [30]. The N50 and N90 of the transcripts were 2356 bp and 1237 bp, respectively, suggesting that the sequence quality was high and could be used for subsequent analyses. Furthermore, more than 95% of transcripts were less than 4 kb in length (Figure 2B). Gene annotation showed that there 97.07% (16,566) of genes were found at least in one database and 7553 transcripts were annotated by all the seven databases used for the analyses (NR, SwissProt, KEGG, KOG, GO, NT and Pfam) (Figure 2C). Finally, a total of 16,380, 14,287, 16,434, 12,984, 10,947, 11,929 and 14,287 genes were annotated in the NR, SwissProt, KEGG, NT, KOG, GO and Pfam databases, respectively (Figure 2C).

### 3.3. Screening of Differentially Expressed Genes (DEGs) and Co-Expression Analysis

To identify genes involved in tapetum degradation and independent microspore development, we analyzed the DEGs among three anther development periods, including Lo 4 cm, Lo 6 cm and Lo 8 cm. Here, DEGs were selected according to their possible functions and fold changes in the expression levels, as well as the screening threshold of padj < 0.05, which were based on the NR annotation database.

A total of 6896 DEGs was screened among the three stages (Figure 3A). In addition, a heat map representation of the expression pattern of the 6896 DEGs, including a hierarchical clustering analysis, can be seen in Figure 3B. The heat map analysis showed that most DEGs among the Lo 4 cm, Lo 6 cm and Lo 8 cm stages showed anti-correlated expression patterns (i.e., they were exactly opposite) (Figure 3B). Furthermore, in total, 4592 DEGs, 2595 DEGs and 4901 DEGs were identified in the comparisons Lo 6 cm vs. Lo 4 cm, Lo 8 cm vs. Lo 6 cm and Lo 8 cm vs. Lo 4 cm, respectively (Figure 3C). The number of up- and down-regulated genes was also counted in each group comparison (Figure 3C). The number of DEGs in the comparison Lo 6 cm vs. Lo 4 cm was almost twice that in Lo 8 cm vs. Lo 6 cm, indicating that the transition from the Lo 4 cm to Lo 6 cm stages was important for tapetum degradation and microspore development, which is consistent with the results of the cytology observations (Figure 1).

Next, we conducted a co-expression analysis of 6896 DEGs and identified 9 different clusters of gene expression patterns (Figure 3). The DEGs were divided into nine clusters and, within the same cluster, the DEGs had similar expression patterns (Figure 3D; Appendix A). In cluster 1, the DEGs were highly expressed from Lo 4 cm to Lo 6 cm but were down-regulated from Lo 6 cm to Lo 8 cm, where the genes related to plant hormone signal transduction were enriched (Appendix A). In cluster 2, the DEGs were only highly expressed at the Lo 6 cm stage, enriching metabolic genes related to fatty acid degradation, amino acid and energy metabolism, MAPK signal pathway, autophagy and peroxisome (Appendix A). In cluster 3, the DEGs were down-regulated from Lo 4 cm to Lo 6 cm and had low gene expression levels from Lo 6 cm to Lo 8 cm, containing genes related to DNA replication, translation and photosynthesis (Appendix A). In cluster 4, the DEGs were down-regulated at Lo 6 cm and enriched genes were involved in translation. In cluster 5, the DEGs were continuously down-regulated from Lo 4 cm to Lo 8 cm, especially genes that participated in photosynthesis, spliceosome, pentose phosphate pathway, phenylpropanoid biosynthesis and flavonoid biosynthesis (Appendix A). In cluster 6, the DEGs had low expression levels from Lo 4 cm to Lo 6 cm but were highly expressed at Lo 8 cm and only one pathway was enriched, i.e., basal transcription factors (Appendix A). In cluster 7, the DEGs were continuously up-regulated from Lo 4 cm to Lo 8 cm, especially genes involved in oxidative phosphorylation, lysosome, ubiquitin-mediated proteolysis and autophagy (Appendix A). In cluster 8, the DEGs were down-regulated from Lo 4 cm to Lo 6 cm but were up-regulated from Lo 6 cm to Lo 8 cm and genes specific to the regulation of phagosome were enriched (Appendix A). In cluster 9, the DEGs were up-regulated from Lo 4 cm to Lo 6 cm and had high expression levels from Lo 6 cm to Lo 8 cm, with enriched pathways including peroxisome, fatty acid, ether lipid metabolism, citrate cycle and oxidative phosphorylation (Appendix A). Based on these co-expression analyses, it can be suggested that there are many genes involved in tapetum degradation and independent microspore development; regulating protein biosynthesis, transport and degradation; lipid acid, plant hormones and secondary metabolism; and further controlling cell division, cell death and cell differentiation.

### 3.4. GO Enrichment and KEGG Pathway Analysis

We performed a GO (gene ontology) enrichment analysis and identified that DEGs were enriched within three major categories, namely, biological process (BP), cellular component (CC) and molecular function (MF) [31]. It was worth mentioning that we thought anther development was a continuous process, so we did not analyze the DEGs in the comparison between Lo 8 cm and Lo 4cm. Additionally, we used the volcano map to visualize the distribution of DEGs in the comparisons Lo 6 cm vs. Lo 4 cm and Lo 8 cm vs. Lo 6 cm (Figure 4C; Appendix A).

In the comparison Lo 6 cm vs. Lo 4 cm, BP category contained 14 annotations, of which the primary terms were: single-organism process and single-organism metabolic process (Figure 4A). The MF category contained 20 annotations, with the main ones being catalytic activity, transferase activity and binding functions. However, few DEGs were enriched in the CC category (Figure 4A). Compared with DEGs in Lo 4 cm vs. Lo 6 cm, fewer genes could be enriched in the comparison between Lo 6 cm and Lo 8 cm and DEGs were only enriched in the BP and MF categories. The BP category contained nine annotations, with the main ones being metabolic process, single-organism process and single-organism metabolic process, which is consistent with the results of Lo 6 cm vs. Lo 4 cm (Figure 4B). The MF category also contained nine annotations, including the primary terms catalytic activity, oxidoreductase activity and transporter activity (Figure 4B). The above GO enrichment analysis suggests that the occurrence of genes related to metabolism and enzyme catalysis could affect tapetum degradation and microspore development.

Congruently, we also analyzed pathways enrichment of DEGs in the KEGG (Kyoto Encyclopedia of Genes and Genomes) database, a resource for understanding high-level functions and utilities of the biological system [32]. Here, we listed the top 20 KEGG pathways enrichment results obtained from the comparisons of the Lo 6 cm vs. Lo 4 cm and Lo 8 cm vs. Lo 6 cm stages (Figure 5A,B). From our data, six pathways, ‘alpha-Linolenic acid metabolism’, ‘Pyruvate metabolism’, ‘Glycolysis/Gluconeogenesis’, ‘Phenylpropanoid biosynthesis’, ‘Fatty acid degradation’, ‘beta-Alanine metabolism’ and ‘Carbon fixation in photosynthetic organisms’ that were related to substance biosynthesis and metabolism, such as amino acids and lipids, were significantly enriched in both comparisons, which is consistent with the GO enrichment results (Figure 4 and Figure 5). In addition to common pathways, we also found that, during the period from the Lo 4 cm to Lo 6 cm stages, 47 genes were enriched in the TCA cycle, an essential pathway to provide energy for the organism (Figure 5A) [33]. In the comparison of the Lo 8 cm vs. Lo 6 cm stages, a total of 32 DEGs were enriched in the ‘Plant hormone signal transduction’ pathway (Figure 5B). Considering that plant endogenous hormones such as GA and IAA change rapidly during tapetum degradation [34], it may play conserved functions in regulating the anther development in lily.

### 3.5. Analysis of Transcription Factors Related to Tapetum Degradation and Microspore Development

Transcription factors (TFs) can regulate the expression of genes associated with anther development [35]. For example, an important MADS-box transcription factor, AGAMOUS (AG), is able to determine the emergence of stamen primordia. The *ag* mutant cannot form the tapetum and endothelium layer normally [36,37,38]. Several MYB proteins are also involved in tapetum development. In Arabidopsis, *AtMYB33* is involved in tapetum differentiation and *AtMYB103*/*80* can participate in the late stage of tapetum development [39,40,41]. In rice, two rice MYB proteins, OsCSA2 and OsCSA, control male sterility by affecting the sugar partitioning in rice anthers [42,43]. Therefore, we searched for differentially expressed TFs in our transcriptome data.

We first conducted expression profiling of all transcription factors. A total of 1192 transcription factors (TFs) were identified and then classified into 25 families (Figure 6A). Here, we put the TF families with relatively few members under the category of ‘others’ (Figure 6A). In addition, a heatmap representation of TF expression profiles and hierarchical clustering analysis is shown in Figure 6B. The heatmap shows that 1192 TFs were expressed differentially among the Lo 4 cm, Lo 6 cm and Lo 8 cm stages (Figure 6B).

Previous studies have shown that MADS-box transcription factors (such as AG) and MYB transcription factors (such as AtMYB33, AtMYB103/80, OsCSA) play an important role in the development of anthers [37,44]. In addition, bHLH transcription factors, DISFUNCTIONAL TAPETUM1(DYT1) and ABORTED MICROSPORES (AMS) from Arabidopsis are also important for tapetal and microspore development [9,45,46]. The tapetum differentiation and callose degradation processes are abnormal in the *dyt1* mutant [47]. In rice, TAPETUM DEGENERATION RETARDATION (TDR), which is the ortholog of AMS, can interact with ETERNAL TAPETUM1 (EAT1) and regulates tapetum development [48,49]. It has been shown that the PHD-finger genes, such as the Arabidopsis *MALE STERILITY 1* (*MS1*) and its rice ortholog *PERSISTENT TAPETAL CELL1* (*OsPTC1*) gene, are critical for pollen formation in plants [38,50]. The homology of these genes (*AtAMS* and *OsTDR*; *AtMS1* and *OsPTC1*) suggests a conserved switch in the regulation of anther development in both dicots and monocots. Therefore, we shed light on these four transcription factor families, MADS-box, MYB, bHLH and PHD, in the transcriptome data. Additionally, the NAC transcription factors, NAC SECONDARY WALL THICKENING PROMOTING FACTOR 1 and 2 (NST1 and NST2), are required for the anther dehiscence in Arabidopsis [51,52]. Similarly, in wheat, 11 TabZIP genes have been identified to be predominantly expressed in anthers, based on whole wheat genome sequencing [53]. These results suggest that members of the NAC family and bZIP family may also be involved in the process of tapetum and microspores development. Hence, we also took the NAC and bZIP families into account to find differentially expressed transcription factors.

From Figure 6A, these six families NAC, MYB, bZIP, bHLH, MADS and PHD, were found to have 23, 61, 52, 48, 39 and 36 members (259/1192), respectively, accounting for 1.93%, 5.12%, 4.36%, 4.03%, 3.27% and 3.02% of all TFs identified. Next, we used BLAST to obtain annotation information of these 259 transcription factors and combined information with the Swiss-Prot and Pfam databases for further screening, to obtain 45 candidate TFs to focus our further analyses on (Figure 7). The heatmap analysis showed the number of members of each family and their differential expression at different stages of anther development, respectively (Figure 7). Among the six TF families, the MADS-box family accounted for the largest percentage, containing nine TFs (Figure 7E). Two *AGAMOUS-LIKE* genes, *AGL14* and *AGL19,* with higher expression levels may function at the late stages of pollen formation (Figure 7E). The NAC, MYB, bZIP and bHLH families had the same number of members, all of which had eight TFs (Figure 7A–D). In addition, the PHD family had only four TFs (Figure 7F). Among them, two *MYB33* genes participated in the tapetum differentiation (Figure 7B). In rice, *ABSCISIC ACID-INSENSITIVE 5* (*OsABI5*), which belong to the bZIP family, are involved in the regulation of pollen maturation. The rice *abi5* mutant exhibits abnormal pollens causing low fertility rates [54]. We found three DEGs of *ABI5* in the transcriptome and they may regulate pollen development in lily (Figure 7C). Considering their differential expression during anther development, it is possible that these 45 transcription factors are involved in the tapetum degradation and microspore development in the lily cultivar ‘Siberia’.

### 3.6. Identification of DEGs Related to Tapetum Degradation and Microspore Development

Based on the gene annotations, we checked a total of 6896 DEGs one by one, as well as the expression pattern of their homologous genes in anther using Arabidopsis ‘eFP browser’. We finally identified 59 DEGs related to tapetum degradation and microspore development and divided them into 5 groups—cell-cycle- and division-related genes, cell-wall-formation-related genes, transcription-factor-related genes, leucine-rich repeat receptor-like kinase (LRR-RLK)-related genes and plant-hormone-synthesis- and signal-transduction-related genes (Figure 8A–E). All the expression patterns and annotations of the DEGs are shown in the heatmap in Figure 8.

The tapetum plays a significant role in the development of anthers. When the microspores complete meiosis, the tapetum degrades the callose to release the microspores and, when the pollen grains enter the mitosis, the tapetum layer is further degraded to form a pollen coat [5,41]. Therefore, the degradation of the tapetum is accompanied by cell division and expansion; four DEGs were thus selected from the transcriptome database. Two cyclin-related DEGs were expressed more in the early stages, while the rest of the genes were expressed more in the Lo 6 cm and Lo 8 cm stages (Figure 8A). These genes may have an effect on mitosis and meiosis during microspore and tapetum development.

Except for the fact that the tapetum is generally involved in the cell division process, it has a vital effect on the formation of the pollen wall. The tapetum can synthesize the precursor material of the outer wall of pollen and release the required material for the inner wall of pollen after being degraded [55,56,57]. Given the relationship between the tapetum and the pollen wall, we identified seven DEGs that were related to cell-wall synthesis (Figure 8B). Three genes were highly expressed in the Lo 4 cm stage and may be involved in the process of precursors synthesis of the outer pollen wall. The other five genes were highly expressed in a later stage, which may be involved in the formation of the inner pollen wall (Figure 8B).

Previous research showed that transcription factors and leucine-rich repeat receptor-like kinases (LRR-RLKs) play important roles in anther development [35,58]. Thus, we identified 11 transcription factors (Figure 8C). The heat map shows that, except for *AGL19*, the rest of them were up-regulated in the period from Lo 4 cm to Lo 6 cm and could be used for subsequent analyses. It has been reported that these genes, *LRR-RLK*s, *EMS1*/*EXS*, *BAM1*/*2*, *SERK1*/*2* and *RPK2,* can affect tapetum formation and degradation [59,60,61]. For example, the *bam1*/*2* double mutant lacks endothelium and tapetum [62,63]. In the process of searching for DEGs, we found five *BAM1*, which were able to encode LRR-RLK proteins (Figure 8D). All these genes were highly expressed in the Lo 4 cm and Lo 6 cm stages, suggesting that they were possibly involved in tapetum development (Figure 8D). Meanwhile, we also considered that tapetum degradation and microspore development may also be regulated by plant hormones [34,64,65], so we found seven genes with high expression levels in the Lo 4 cm and Lo 6 cm stages (Figure 8E). They were involved in the synthesis and signal transduction of IAA, ABA, JA and GA. For example, one DEG encodes NCED (9-cis-epoxycarotenoid dioxygenase), a rate-limiting enzyme in the ABA biosynthesis, and GID1C, an important receptor in the GA signaling pathway. These DEGs were candidates for tapetum degradation in lily.

### 3.7. Differential Gene Expression Validation by qRT-PCR Analysis

To further verify the transcriptomic analysis, 16 DEGs were selected for qRT-PCR validation. These genes included transcription factors related to anther development and callose degradation, genes related to microspore maturation and genes related to plant hormone signaling. The results showed that the expression patterns of these DEGs were similar to the changes in FPKM values in the transcriptome data (Figure 9), thus supporting that our transcriptomic data and analyses are of high quality.

## 4. Discussion

Pollen is produced from the anthers in the stamens and reaches the pistils for fertilization [66]. The development of pollen goes through the stages of archesporial cells, sporogenous cells, dyads, tetrads and microspores; then, it gradually develops into mature pollen grains [67]. Mature pollen grains play a significant role in sexual reproduction. However, in some ornamental plants, such as lily flowers, the presence of a large number of pollen grains may cause allergies in some people and hinder the breeding process [2]. The tapetum is a special cell layer, the closest one to the anther chamber. It can provide a series of enzymes and nutrients for microspore development and secretes β-1,3-glucanase to dissolve the callose for microspore releasing [68,69]. The microspores further develop into mature pollen. Moreover, the degradation of the tapetum itself forms a pollen coating, which is necessary for pollen germination [70]. Therefore, the tapetum is crucial for pollen development and maturation.

In this study, we used a popular lily cultivar, ‘Siberia’, as plant material. We continuously observed the development of lily anthers and found that anthers from 4 cm, 6 cm and 8 cm (Lo 4 cm, Lo 6 cm and Lo 8 cm) buds were the key stages for tapetum degradation and microspore development. Then, anthers from Lo 4 cm, Lo 6 cm and Lo 8 cm were selected for subsequent transcriptome sequencing. We finally identified several genes possibly related to tapetum degradation and microspore development by a series of comparative analyses. By identifying the key genes regulating tapetum development and degradation, we can better understand the process of pollen development and contribute to the production of new lily varieties without pollen contamination by genetic engineering.

We first clarified the development process of the tapetum and microspores by paraffin-sectioning the anthers of lily buds at different lengths (Figure 1). It is reported that Sanders et al. divided the development of Arabidopsis stamens into 14 stages [67]. As Sanders et al. described, microspore mother cells differentiated and formed in the stages 1–5. The tetrads formed and were surrounded by callose in the stages 6–7, which corresponds to our observation of Lo 4 cm (Figure 1C). The microspores are released from the tetrad and develop further in the 8–11 period, which corresponds to Lo 6 cm (Figure 1D). Then, the pollen grains gradually mature with the tapetum degradation and are released from the anther in the 12–14 period, which is similar to what we observed in the Lo 8 cm stage (Figure 1E). Therefore, we thought that there was a certain similarity in tapetum and micropore development between Lily and Arabidopsis.

The degradation of tapetum and the development of microspores are regulated by strict cell fate determination and morphogenesis mechanisms [71,72]. Consequently, we found four genes, which were possibly involved in the cell cycle and division (Figure 8A). These genes may have an influence on the fate of tapetum cells and microspores. In Arabidopsis, TAPETUM DETERMINANT1 (TPD1) is a potential small protein-ligand of the EXCESS MICROSPOROCYTES1 (EMS1) LRR-RLK and the TPD1-EMS1 signal module is specifically required for anther cell differentiation during sexual reproduction [73,74]. Overexpression of *TPD1* increases the number of carpel cells and delays the degradation of tapetum cells [74]. Its rice homolog, *OsTPD1-like (OsTDL1A),* binds to the receptor kinase *MULTIPLE SPOROCYTES1 (MSP1)* and is also required for cell proliferation in early pollen development [75,76]. In addition, studies have shown that changes in the expression level of *TPD1* not only affected the endogenous auxin response but also enhanced the expression of cyclin genes *CYCD3;3 and CYCA2;3*, suggesting that *TPD1* functions by affecting the auxin response pathway and cell-cycle process [73]. Similarly, we found one DEG, annotated as *TPD1,* and two cyclin genes, *CYCA2;1* and *CYCD4;1,* with different expression levels at the Lo 4 cm, Lo 6 cm and Lo 8 cm stages (Figure 8A). This suggests that they may have a conserved function in lily tapetum and microspores development (Figure 8A). Apart from the *TPD1* and cyclin genes, two DEGs, which were respectively aligned to *EXO70A1* and *DAR1* (*DA1-RELATED 1*), were identified to be expressed more in the Lo 6 cm and Lo 8 cm stages (Figure 8A). Previously, *EXO70A1* has been shown to be required to control pollen hydration and pollen tube development and *DAR1* can regulate endoreduplication in leaf epidermal tissue [77,78]. Through the eFP browser, we observed that *DAR1* (AT4G36860) in Arabidopsis was also expressed differentially during anther development (Appendix A). Therefore, these genes are possibly involved in cell fate determination in lily anthers.

The microspore mother cell undergoes meiosis to form microspores, accompanied by vacuolation of tapetum cells [79]. When meiosis is complete, tapetum cells can synthesize sporopollenin and other substances to provide the material basis for the formation of the microspore pollen wall. After the microspores are released, tapetum degradation can provide material and energy for the formation of the pollen wall [41,57]. Considering that the development of tapetum and microspores is closely related to the process of cell-wall formation, we identified seven enzymes possibly involved in these processes (Figure 8B). These genes encode cellulose synthase (*CSLD2*, *CESA1* and *CESA6*), xylose isomerase (*AT5G57655*), pectin-esterase (*PE11*), sucrose synthase (*SUS1*) and beta-1,3-glucanase (*BGL2*), respectively, and have been shown to participate in the cell-wall formation process (Figure 8B) [80,81,82,83]. These seven genes were differentially expressed in the Lo 4 cm, Lo 6 cm and Lo 8 cm stages of lily bud development. We observed the expression levels of these genes in the eFP browser and found that, except for beta-1,3-glucanase-related genes with low expression levels, the other genes were differentially expressed in the anther development of Arabidopsis (Figure 8B and Appendix A). Hence, we thought that these six enzyme-encoding genes had a potential impact on tapetum and microspores development.

LRR-RLKs (leucine-rich repeat receptor-like kinases) are key components in many plant signal-transduction processes and play an important role in plant growth and development, resistance to adversity stress, hormone signal transduction, etc. [84,85,86]. Notably, several LRR-RLKs work together to control anther-cell differentiation. EXCESS MICROSPOROCYTES1 (EMS1)/EXTRA SPOROGENOUS CELLS (EXS) and its small protein-ligand TAPETUM DETERMINANT1 (TPD1), SOMATIC EMBRYOGENESIS RECEPTOR-LIKE KINASES1/2 (SERK1/2) and BARELY ANY MERISTEM1/2 (BAM1/2) directly have a significant impact on the formation of the endothecium, middle layer, tapetum and other anther structures [87,88,89,90]. Besides, a novel factor, RECEPTOR-LIKE PROTEIN KINASE (RPK2), can control tapetum development in the late stage of the anther in Arabidopsis [91]. In rice, *MULTIPLE SPOROCYTES1* (*OsMSP1*), which encodes an LRR-RLK, is closely similar to *AtEXS/EMS1* in structure and function [92]. In all these mutants, the lack of these kinases results in the repression of tapetum formation and development. In our database, we found five DEGs that were annotated as *BAM1* but no other *LRR-RLK*s were found (Figure 8D). They were expressed more in the early stage of the lily anther, which may affect the development of tapetum in lily (Figure 8D). This result also reflects that the regulation mechanism of anther development is somewhat conserved.

The physiological effects of plant hormones are complex and diverse. The dynamic changes in endogenous hormone content and the hormone signal transduction are involved in the various processes of plant growth and development [93,94,95]. Previous research has demonstrated that the development of anthers is regulated by plant hormones. Hirano K. et al. analyzed the endogenous content of plant hormones in different tissues of rice and found that the contents of IAA and GA in mature anthers were significantly higher than those in other tissues [64]. In the *Petunia hybrida* L., it was demonstrated that ABA and IAA were involved in the programmed cell death (PCD) of microsporocytes at the meiosis stage [96]. In addition, JA is also involved in the reproductive process of plants and the phenotype of the JA biosynthetic mutant is profound sporophytic male sterility [97]. Consequently, we paid attention to these four hormones, ABA, GA, IAA and JA, and found seven related DEGs. These genes were either related to hormone synthesis or as hormone receptors involved in hormone signal transduction and may participate in tapetum degradation and micropore development. As previous research has shown, *AUX/IAA* and *ARF (AUXIN RESPONSE FACTOR)* genes play a crucial role in pollen wall formation, tapetum development and anther dehiscence [64,98,99,100,101]. We found two DEGs, *IAA8* and *ARF23*, whose expression levels were high in the Lo 4 cm period, indicating that they may play a regulatory role in the early stage of tapetum and microspore development (Figure 8E). NCED is a key rate-limiting enzyme in the ABA synthesis pathway and PYL8 is an important ABA receptor [102,103]. Two DEGs were identified as *NCED* and *PYL8* in lily, respectively, and they were all highly expressed in the Lo 6 cm stage (Figure 8E). There are five *NCED* homologous genes in Arabidopsis, namely, *NCED2*, *NCED3*, *NCED5*, *NCED6* and *NCED9*, and their expression levels in anthers were low in the eFP browser (Appendix A). Conversely, *PYL8*, which corresponds to AT5G53160 in *Arabidopsis thaliana*, was expressed differently at different stages of anther development (Appendix A), indicating that *PYL8* may regulate the development of tapetum and microspores through the ABA signal transduction pathway. In addition, we also found two DEGs, which were respectively annotated as *GID1C* and *JAR1* (Figure 8E); GID1C is an important receptor of the GA signaling pathway and JAR1 is involved in JA biosynthesis [104,105]. These DEGs were highly expressed in the Lo 6 cm period (Figure 8E). Through the eFP-browser tool, *GID1C* (AT5G27320) showed no differences in the expression of anthers development, but *JAR1* (AT2G46370) was significantly less expressed in mature pollen grains than in other stages (Appendix A), indicating that *JAR1* may have an impact on the development of anthers by affecting the JA synthesis process. In fact, it is still unclear how plant hormones regulate tapetum degradation and microspore development and further research is required to clarify the underlying mechanisms.

In conclusion, we used ‘Siberia’, an important lily cut flower variety, as the experimental material and determined that the period from Lo 4 cm to Lo 8 cm is the key period for the degradation of anther tapetum and the development of microspores. The transcriptome analysis revealed that several DEGs, involved in cell division and expansion, cell-wall morphogenesis, transcription factors, LRR-RLKs, plant hormone biosynthesis and transduction, possibly regulated tapetum degradation and micropore development in lilies. The quantitative real-time PCR of the DEGs in three stages, Lo 4 cm, Lo 6 cm and Lo 8 cm confirmed the transcriptome data. This study helps us better understand the development process of lily anthers and provides valuable candidate genes to be further studied that may make it possible to cultivate new varieties without pollen contamination.

## Figures and Tables

**Figure 1 genes-13-00366-f001:**
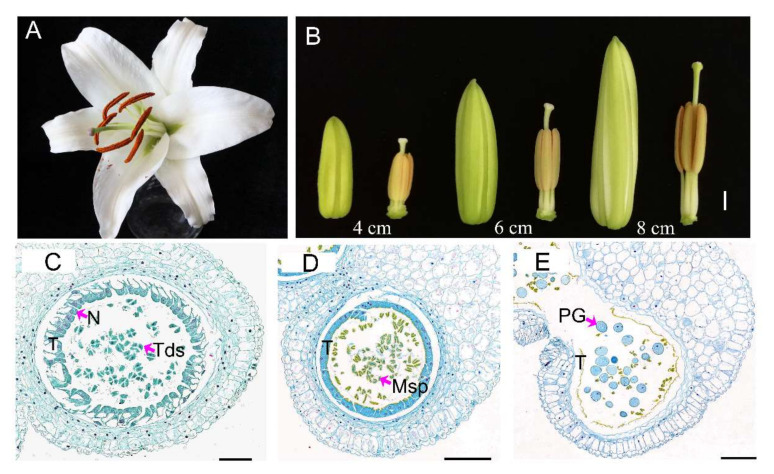
Morphological and histological analyses of lily flower buds. (**A**) Lily flower when pollen grain was released. (**B**) Lily flower buds and anthers at Lo 4 cm, Lo 6 cm and Lo 8 cm stages. The scale bar is 1 cm. (**C**) Anther sections at Lo 4 cm stage. The scale bar is 100 μm. (**D**) Anther sections at Lo 6 cm stage. The scale bar is 200 μm. (**E**) Anther sections at Lo 8 cm stage. The scale bar is 200 μm. T, tapetum; Tds, tetrads; N, nucleus; Msp, microspores; PG, pollen grain.

**Figure 2 genes-13-00366-f002:**
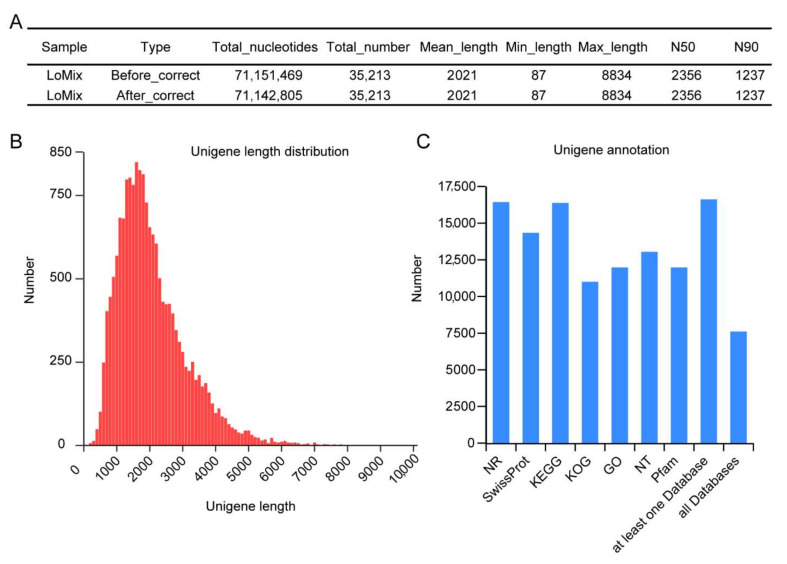
Summary of transcriptome sequencing and gene annotation. (**A**) Summary of transcripts before and after correction. (**B**) Length distribution map of unigenes. (**C**) Statistics of functional annotation in seven databases.

**Figure 3 genes-13-00366-f003:**
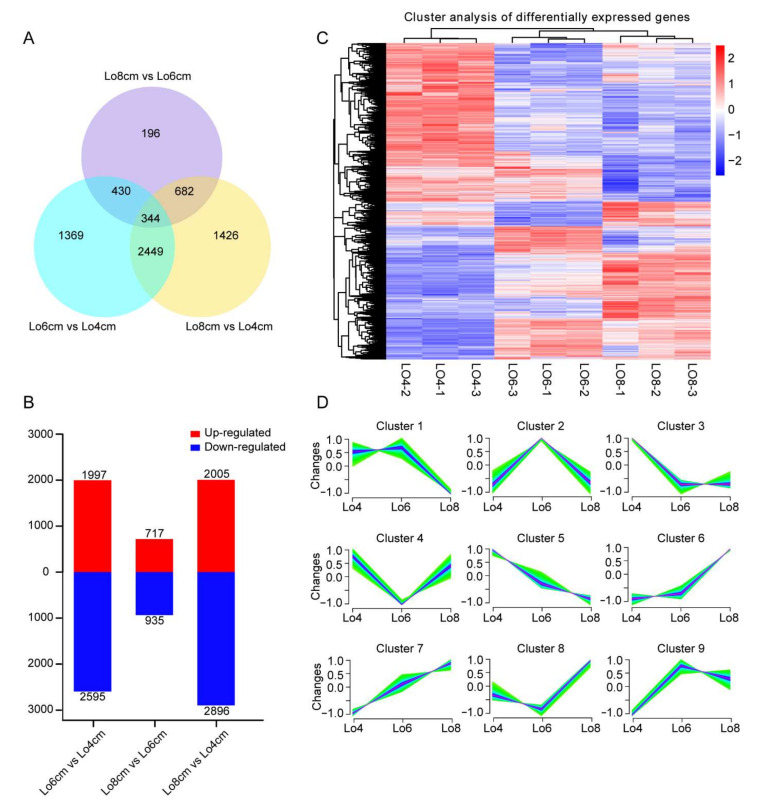
DEGs among Lo 4 cm, Lo 6 cm and Lo 8 cm. (**A**) Venn diagram showing the number of DEGs in different comparisons. (**B**) Numbers of DEGs in pairwise comparisons. (**C**) Heatmap diagrams showing the relative expression levels of DEGs. The color scale on the right from blue to red represents FPKM values from low to high. (**D**) The trend analysis of DEG expression fold-change divided into nine clusters based on expression patterns.

**Figure 4 genes-13-00366-f004:**
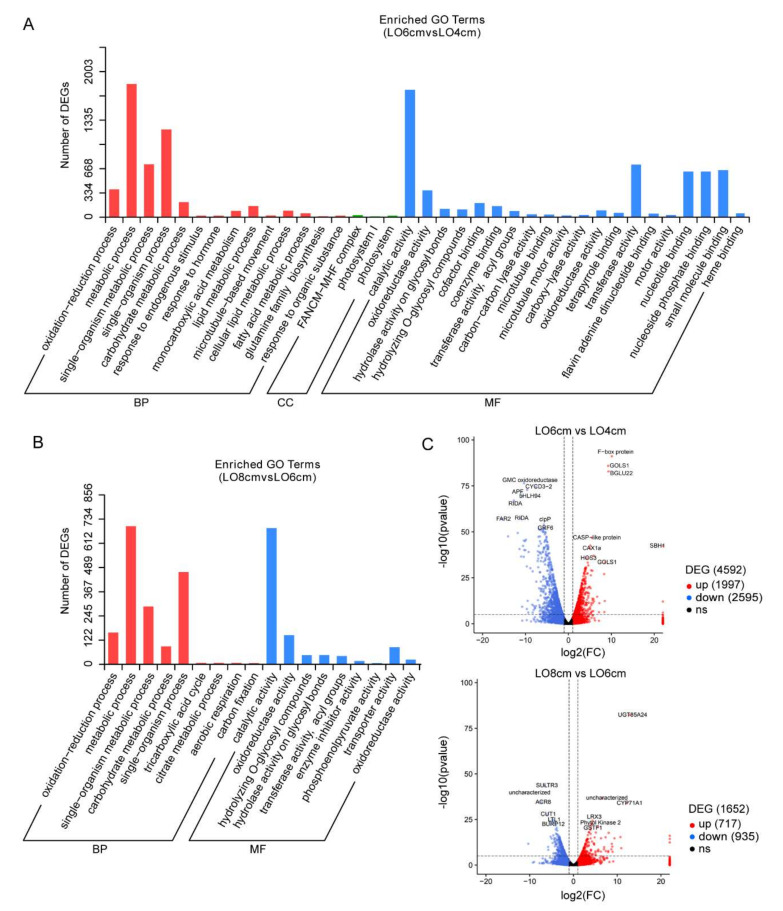
GO classification of DEGs in Lo 6 cm vs. Lo 4 cm and Lo 8 cm vs. Lo 6 cm stages. (**A**) Enhanced GO terms in Lo 6 cm vs. Lo 4 cm. (**B**) Enhanced GO terms in Lo 8 cm vs. Lo 6 cm. (**C**) Volcano chart showing the distribution of DEGs for each comparison. The x-axis represents the log_2_FC (FoldChange) value, the y-axis the −log10 *p* value and the dotted line represents the threshold line of the DEGs screening criteria. The red dots and blue dots represent up- and down-regulated genes, respectively, and genes with no significant differential expression are represented by black dots. BP, biological process; MF, molecular function; CC, cellular component.

**Figure 5 genes-13-00366-f005:**
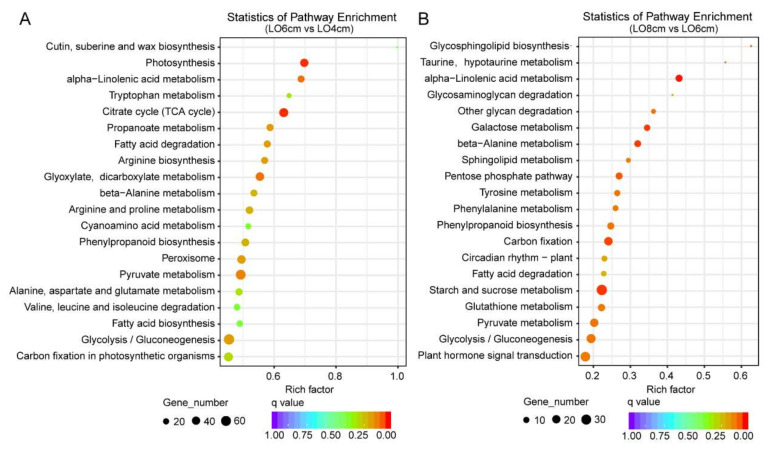
The top 20 enriched KEGG pathways among the DEGs. (**A**) Enriched KEGG pathways in comparison Lo 6 cm vs. Lo 4 cm. (**B**) Enriched KEGG pathways in Lo 8 cm vs. Lo 6 cm stages. The y-axis presents the different KEGG pathways and the x-axis presents the enrichment factor. Dot colors represent the q-value and dot sizes represent gene numbers enriched in the corresponding pathway.

**Figure 6 genes-13-00366-f006:**
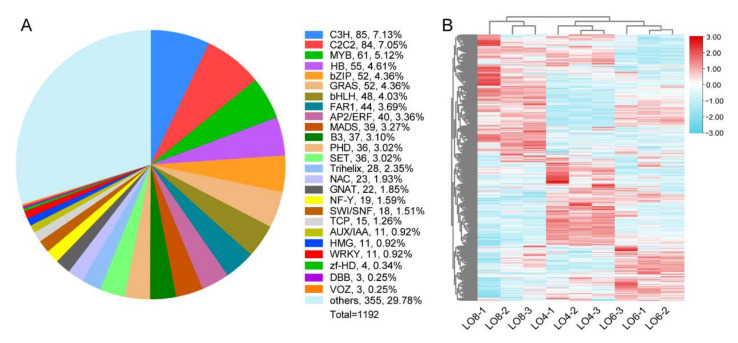
The overall expression of TFs in DEGs. (**A**) Numbers of total differentially expressed transcription factors. (**B**) Heatmap diagrams showing the relative expression levels of differentially expressed transcription factors. The color scale on the right from blue to red represents FPKM values from low to high.

**Figure 7 genes-13-00366-f007:**
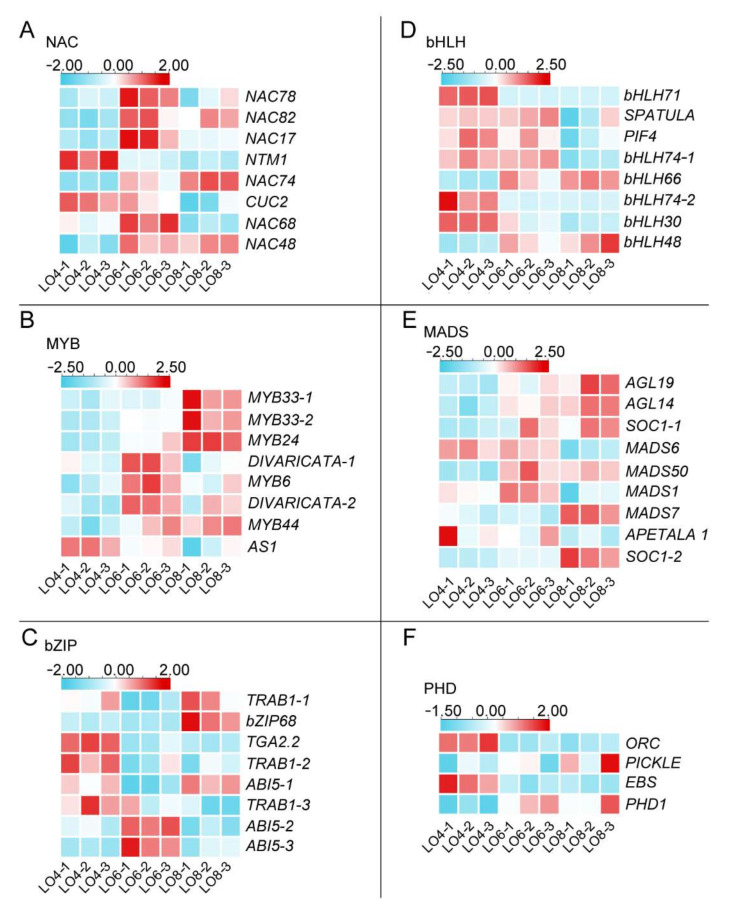
Heat map diagrams showing the relative expression levels of TFs possibly associated with the development of tapetum and microspore. (**A**) NAC (NAM/ATAF/CUC) family member. (**B**) MYB family members. (**C**) bZIP (basic region/leucine zipper) family members. (**D**) bHLH (basic helix-loop-helix) family members. (**E**) MADS-box family members. (**F**) PHD (plant homeodomain)-finger protein family members. The color scale on the top from blue to red represents FPKM values from low to high.

**Figure 8 genes-13-00366-f008:**
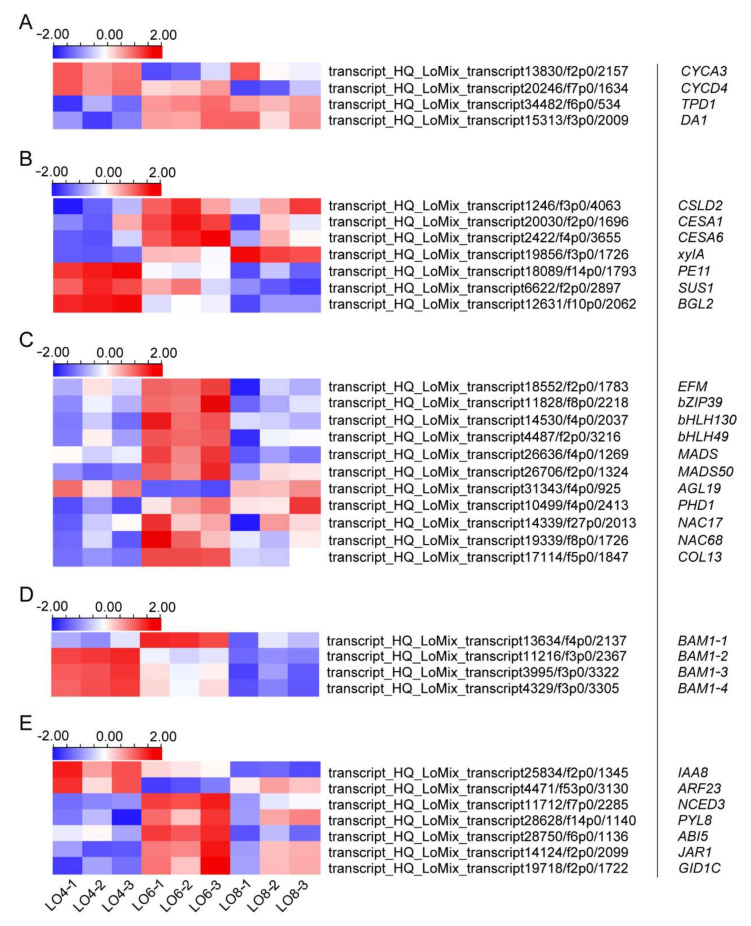
The heatmap expression representation of candidate DEGs involved in the tapetum degradation and the development of microspores. (**A**) Cell-cycle- and division-related genes. (**B**) Cell-wall-formation-related genes. (**C**) Transcription-factor-related genes. (**D**) Leucine-rich repeat receptor-like kinase (LRR-RLK)-related genes. (**E**) Plant-hormone-synthesis- and signal-transduction-related genes. The color scale on the top from blue to red represents FPKM values from low to high.

**Figure 9 genes-13-00366-f009:**
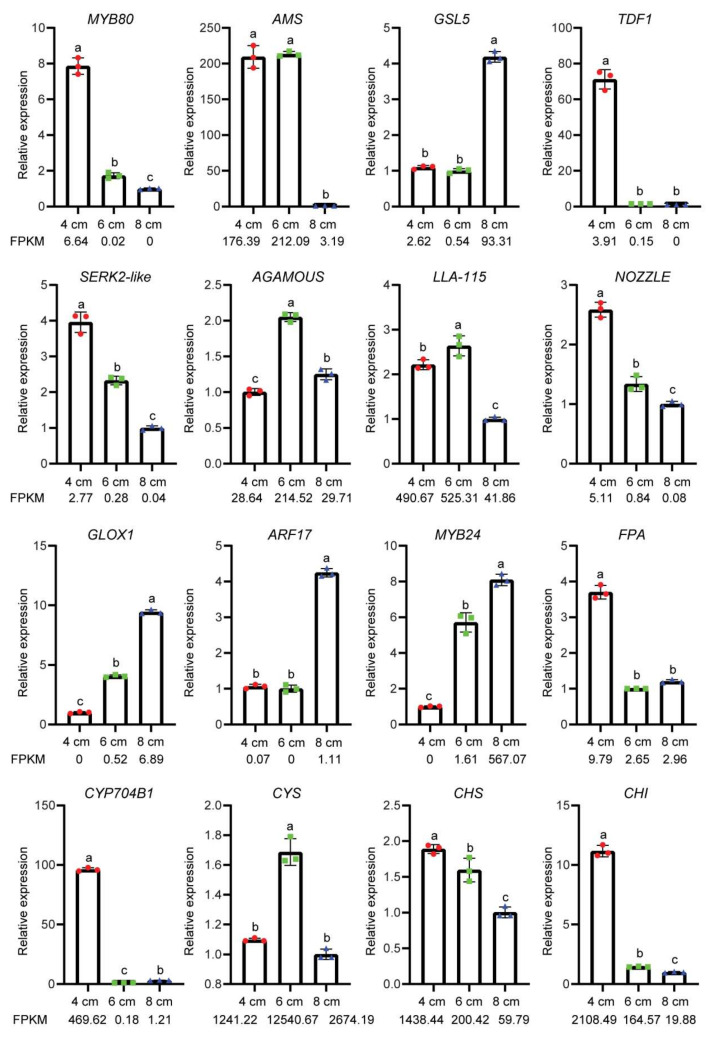
Relative expression analysis of the candidate DEGs. The values of qRT-PCR were determined using the 2^−ΔΔCT^ method, the lowercase letters “a”, “b” and “c” represent statistically significant differences calculated by an ANOVA and post hoc Tukey’s HSD (*p* < 0.05). FPKM values, listed below the x-axis, are the counts from RNA-seq data.

## Data Availability

The transcriptome data are available at the National Genomics Data Center of China (project PRJCA008073; data access link: https://ngdc.cncb.ac.cn/gsa/browse/CRA005948, accessed on 23 June 2021). The codes used for data analysis methods are reported in Appendix A.

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
