# Peer review of "Transcriptomics-Based Identification of Genes Related to Tapetum Degradation and Microspore Development in Lily"

_genes, 2022, doi:10.3390/genes13020366_

Round 1

Reviewer 1 Report

The paper of Sui et al. aimed to identify genes that are relate d to tapetum degradation and microspore development in Lilium. Also, figures in this paper were not properly placed as to where they should be. Figures are placed right after you mention them. The authors must also observe the proper way of writing a gene name, it should be italicized. There are also unnecessary changes with the font size (see line 422) and things like “Error! Reference source not found.B, E” across the manuscript. In addition to this, several stated claims are not acceptable for a scientific publication. For example, line 262 “The patterns in clusters 4, 262 6, and 8 indicated that the DEGs may be involved in the later stage of anther development.” Better look for studies that would support the claim, or at least related to this phenomena. There are a lot of this reasoning across the whole manuscript. Saying maybe in a sentence makes your result a hunch when in fact you did the experiments. 

Author Response

Reviewer 1

The paper of Sui et al. aimed to identify genes that are related to tapetum degradation and microspore development in Lilium. Also, figures in this paper were not properly placed as to where they should be. Figures are placed right after you mention them.

>We have arranged figures based on the journal requirements.

The authors must also observe the proper way of writing a gene name, it should be italicized.

> We have italicized all genes throughout the manuscript including figures.

There are also unnecessary changes with the font size (see line 422) and things like “Error! Reference source not found. B, E” across the manuscript.

> Sorry for this inconvenience. We transferred the revised manuscript into a PDF version and hopefully, it can be readable.

In addition to this, several stated claims are not acceptable for a scientific publication. For example, line 262 “The patterns in clusters 4, 262 6, and 8 indicated that the DEGs may be involved in the later stage of anther development.” Better look for studies that would support the claim or relate to this phenomenon.

>We have rewritten the whole paragraph based on your suggestion. See Line 255-278.

There are a lot of this reasoning across the whole manuscript. Saying maybe in a sentence makes your result a hunch when in fact you did the experiments.

> Thanks for your reminder. We have changed some descriptions in the results.

Reviewer 2 Report

The paper "Transcriptomics-based identification of genes related to tapetum degradation and microspore development in Lilium" by Sui and colleagues describes an original transcriptomics study on the lily species during development. While the dataset is original, the manuscript lacks in methodological soundness, data presentation and overall comparison with other monocot species. Below, my major and minor points.

Major points:

- It is absolutely necessary to compare the results with transcriptomics data deriving from developmental studies in other plants. A prime example would be rice (model plant for monocots), for which many studies and publicly available information exist.

- An sample cluster analysis (e.g. PCA) is required, to show whether the three replicates within each Lo group are in agreement with each other.

- The authors perform extensive pathway enrichment analysis, but do not show nor focus on the top DEGs (differentially expressed genes), except when to confirm their expression with RT-PCR (Figure 9). The authors should definitely show the top 20 DEGs (e.g. 10 most up regulated and 10 most downregulated in each contrast) in the form they prefer. A very popular visualization of differential signatures is for example volcano plots, labeling the most differentially expressed genes. This would help the reader observe agreements (or lack of them) between RNA-Seq and RT-PCR.

- Figure 9 lacks a proper legend: are the genes shown from RT-PCR (as the general flow of the paper seems to point at) or from RNA-Seq? The text says that values are similar, but the figure reports only results from one of the two techniques. It would be appropriate to show results from both.

- There is no mention on the method used to calculate DEGs, which shoul ideally be located between paragraphs 2.6 (quantification of transcripts) and the next paragraphs (investigating DEGs).

- I could not find the data. Online search for the code "PRJCA008073" provided by the authors provides no hits. A search on the National Genomics Data Center of China for the same code provides this link: https://ngdc.cncb.ac.cn/search/?dbId=&q=PRJCA008073 but doesn not allow access to raw data besides a general description of the files. I suggest the authors should upload their samples (in FASTQ or BAM format, if the second also providing a reference genome id) in an internationally-accessible database (GEO, EBI, DDBJ) or provide the raw data on a permanent, stable, fast public link

- My previous point of sharing data extends to the code used to align reads and perform the statistical analysis and plotting used in the paper. The full pipeline should be fully reproducible.

- It is unclear, from the methods section, why the authors did not use the reference lily genome (or a close cultivar/species, such as the one released in 2020 by the Tang lab: https://pubmed.ncbi.nlm.nih.gov/31853069/) to perform read alignment and transcript quanitifcation

- The authors use the scientific term "Lilium" in the title, and nowhere else in the text, where they use the common term "Lily". The authors should be consistent, or at least define once the species they are studying in the text, in scientific naming (e.g. Lilium pensylvanicum), besides the mention of the Siberia cultivar name.

Minor points:

- Line 34: missing citation for statement about user preference for single lilies worldwide.

- Line 160: a sentence "Error! Reference source not found" is inserted in the methods sentence for read alignment.

- Line 182: another sentence "Error!" is inserted in the text

- Line 190, 192, 197... And more: "Error!" words are inserted in many positions of the paper, suggesting a bug in the citation managing software used to write the manuscript

- More typos and wording require an extensive English proofreading (e.g. "proess", "cello layer" in the abstract).

Author Response

Reviewer 2

The paper "Transcriptomics-based identification of genes related to tapetum degradation and microspore development in Lilium" by Sui and colleagues describes an original transcriptomics study on the lily species during development. While the dataset is original, the manuscript lacks in methodological soundness, data presentation and overall comparison with other monocot species. Below, my major and minor points.

Major points:

- It is absolutely necessary to compare the results with transcriptomics data deriving from developmental studies in other plants. A prime example would be rice (model plant for monocots), for which many studies and publicly available information exist.

> We have compared some genes with the orthologues in rice and added references in the manuscript based on your suggestion. See Line 344, 359-366, 383-390, 512 and 557.

- An sample cluster analysis (e.g. PCA) is required, to show whether the three replicates within each Lo group are in agreement with each other.

>We have added the PC2 results as Figure S1 based on your suggestion. The data shows a good consistency within the three replicates and diversities among the three different stages. See Line 218 and Figure S1.

- The authors perform extensive pathway enrichment analysis, but do not show nor focus on the top DEGs (differentially expressed genes), except when to confirm their expression with RT-PCR (Figure 9). The authors should definitely show the top 20 DEGs (e.g. 10 most up regulated and 10 most downregulated in each contrast) in the form they prefer. A very popular visualization of differential signatures is for example volcano plots, labeling the most differentially expressed genes. This would help the reader observe agreements (or lack of them) between RNA-Seq and RT-PCR.

> We added this result in Figure 4C as you suggested in the volcano plots.

- Figure 9 lacks a proper legend: are the genes shown from RT-PCR (as the general flow of the paper seems to point at) or from RNA-Seq? The text says that values are similar, but the figure reports only results from one of the two techniques. It would be appropriate to show results from both.

> We have added the description in the figure legend as ‘FPKM values, listed below the x-axis, were the counts from RNA-seq data.’ in order to make it clear. The charts are generated based on the qRT-PCR data with three biological replicates. Actually, our data showed both results (RNA-seq and qRT-PCR).

- There is no mention on the method used to calculate DEGs, which should ideally be located between paragraphs 2.6 (quantification of transcripts) and the next paragraphs (investigating DEGs).

>We have added the related description in the methods. ‘DEGs were identified using DESeq2 package with the threshold of logarithm transformation of fold change larger than 1 and p-value < 0.05’. See Line 167, Table S3, and Table S4.

- I could not find the data. Online search for the code "PRJCA008073" provided by the authors provides no hits. A search on the National Genomics Data Center of China for the same code provides this link: https://ngdc.cncb.ac.cn/search/?dbId=&q=PRJCA008073 but doesn not allow access to raw data besides a general description of the files. I suggest the authors should upload their samples (in FASTQ or BAM format, if the second also providing a reference genome id) in an internationally-accessible database (GEO, EBI, DDBJ) or provide the raw data on a permanent, stable, fast public link

>Here is the data link: https://ngdc.cncb.ac.cn/gsa/browse/CRA005948. We clarify in Line 619.

- My previous point of sharing data extends to the code used to align reads and perform the statistical analysis and plotting used in the paper. The full pipeline should be fully reproducible.

>We submitted the code that we used in supplementary method 1.

- It is unclear, from the methods section, why the authors did not use the reference lily genome (or a close cultivar/species, such as the one released in 2020 by the Tang lab: https://pubmed.ncbi.nlm.nih.gov/31853069/) to perform read alignment and transcript quanitifcation

>The reference you mentioned (Tang et al., 2020) sequences the species of water lily, a member of the family Nymphaeaceae. However, lily belongs to the Liliaceae family. These two species are different species. Lily’s genome is around 60 Gb, and currently, there is no public genomic resource.

- The authors use the scientific term "Lilium" in the title, and nowhere else in the text, where they use the common term "Lily". The authors should be consistent, or at least define once the species they are studying in the text, in scientific naming (e.g. Lilium pensylvanicum), besides the mention of the Siberia cultivar name.

> We changed ‘Lilium’ into ‘lily’ in the title and made them consistent in the text.

Minor points:

- Line 34: missing citation for a statement about user preference for single lilies worldwide.

> We added the references. See Line 36.

- Line 160: a sentence "Error! Reference source not found" is inserted in the methods sentence for read alignment.

> Sorry for this inconvenience. We transferred the revised manuscript into a PDF version and hopefully, it can be readable.

- Line 182: another sentence "Error!" is inserted in the text

> I could not find these bugs on my computer. I will upload the PDF version of the manuscript along with this submission.

- Line 190, 192, 197... And more: "Error!" words are inserted in many positions of the paper, suggesting a bug in the citation managing software used to write the manuscript

> I could not find these bugs on my computer. I submitted the PDF version of the manuscript in re-submission.

- More typos and wording require an extensive English proofreading (e.g. "proess", "cello layer" in the abstract).

> We corrected all errors throughout the manuscript.

Round 2

Reviewer 1 Report

The authors significantly improved the manuscript.

Reviewer 2 Report

The authors replied to all my points